# Recent Progress on the Adsorption of Heavy Metal Ions Pb(II) and Cu(II) from Wastewater

**DOI:** 10.3390/nano14121037

**Published:** 2024-06-16

**Authors:** Dikang Fan, Yang Peng, Xi He, Jing Ouyang, Liangjie Fu, Huaming Yang

**Affiliations:** 1School of Minerals Processing and Bioengineering, Central South University, Changsha 410083, China; 215611020@csu.edu.cn (D.F.); jingouyang@csu.edu.cn (J.O.); hmyang@csu.edu.cn (H.Y.); 2Engineering Research Center of Nano-Geomaterials of Ministry of Education, Faculty of Materials Science and Chemistry, China University of Geosciences, Wuhan 430074, China; yang0519@cug.edu.cn; 3Key Laboratory of Functional Geomaterials in China Nonmetallic Minerals Industry, China University of Geosciences, Wuhan 430074, China; 4Changsha Industrial Technology Research Institute (Environmental Protection) Co., Ltd., Changsha 410083, China; hexi@kaitiangroup.com; 5Aerospace Kaitian Environmental Technology Co., Ltd., Changsha 410083, China

**Keywords:** heavy metal adsorption, copper, lead, wastewater, adsorbent, review

## Abstract

With the processes of industrialization and urbanization, heavy metal ion pollution has become a thorny problem in water systems. Among the various technologies developed for the removal of heavy metal ions, the adsorption method is widely studied by researchers and various nanomaterials with good adsorption performances have been prepared during the past decades. In this paper, a variety of novel nanomaterials with excellent adsorption performances for Pb(II) and Cu(II) reported in recent years are reviewed, such as carbon-based materials, clay mineral materials, zero-valent iron and their derivatives, MOFs, nanocomposites, etc. The novel nanomaterials with extremely high adsorption capacity, selectivity and particular nanostructures are summarized and introduced, along with their advantages and disadvantages. And, some future research priorities for the treatment of wastewater are also prospected.

## 1. Introduction

Water is an essential element for the birth and maintenance of life, and it has been a top priority in human history since ancient times. However, the pollution of heavy metal ions in water systems has been a serious problem caused by the continuous development of industrialization and urbanization [1,2], although their presence in water is not so obvious [3]. The wastewater from industrial production is the main source of heavy metal ions. During the past half-century, people realized that the accumulation of PPb levels of heavy metal ions can cause serious diseases in the human body and the amount of heavy metal is gradually enriched in the biological body along the food chain with irrigation, drinking water and other ways [4,5,6]. 

Nowadays, many countries have set strict standards for heavy metal ion pollution in water bodies for ecological environment protection. According to the GB3838-2002 standards in China [7], the concentration of Pb, Cu, Zn, Cd and Cr in surface water environments should be less than 0.01, 0.01, 0.05, 0.001 and 0.01 mg/L, respectively. In the water system, heavy metal pollution mainly comes from the following sources: (1) Exhaust gas from industry and automobiles, wastewater from mining and industry, waste residue and waste materials. (2) Pesticides, fertilizers and additives used in agriculture. (3) Domestic waste, medical waste, discarded appliances, etc. [8]. 

Pb and Cu are widely used in the chemical industry, and their pollution degree is particularly serious in soils (mines, urban soils and agricultural soils) [9,10,11,12,13] and various water systems (wastewater, surface water, groundwater, rivers, lakes and ponds) [14,15,16,17,18], which has become a very representative problem in heavy metal pollution in China and worldwide. With the fast development of the copper industry in China, it was reported that the concentrations of Pb(II) and Cu(II) in river water around the typical copper mines (Dabaoshan, Dahongshan, Jingchuan, Baiyin and Dexing) in China had increased up to 2.9 and 136 mg/L, respectively, which indicated that the wastewater from mining activities had heavily contaminated the natural water bodies [14]. The concentrations of Pb and Cu were found far beyond the allowable limits, which became the main ecological risk sources in water systems. Table 1 summarizes the sources of Pb(II) and Cu(II) pollution and their allowable concentrations from different institutions, along with their toxicity, hazards and vulnerable populations.

To date, the pollution of Pb(II) and Cu(II) ions in water systems is still serious due to the high cost of the nanomaterials used in various types of water treatment equipment. Thus, in order to control heavy metal pollution in a wider range, adsorption nanomaterials that are low-cost and easy to operate have been one of the most important methods for wastewater treatment. 

Over the past few years, the efficient removal of Pb(II) and Cu(II) ions from wastewaters has attracted much attention and various novel nanomaterials with double the adsorption ability for both ions have been reported [19,20,21,22,23,24,25,26,27,28]; however, a comprehensive review of the relevant works is still lacking. Therefore, this review focuses on the recent progress on novel nanomaterials for the adsorption of Pb(II) and Cu(II) ions, introducing the characteristics of wastewater, the structure and performance of the most used adsorption materials, as well as their adsorption mechanisms and parameters.

## 2. Characteristics and Toxicological Properties of Wastewater Containing Pb(II) and Cu(II)

### 2.1. The Toxicological Effects of Pb(II) and Cu(II)

Lead (Pb) is a highly toxic heavy metal, and it has become a significant health risk for humans and animals. Pb(II) may exist in the forms of Pb^2+^ and PbOH^+^ in water at acid and neutral conditions, Pb_2_OH^3+^ and Pb(OH)_4_^2-^ at higher pH conditions, or organic complexes with various organic compounds. Pb(II) enters the human body and circulates mainly in the form of a glycerophosphate protein complex and lead ion. Most Pb(II) is stored in the bones and no symptoms of poisoning occur at this time. Symptoms of Pb(II) poisoning occur when the concentration of Pb(II) is too high in the internal organs, soft tissues and blood. The poisoning mechanism of Pb(II) in the human body is to inhibit the enzyme containing sulfhydryl in cells, which harms the biochemical and physiological functions of human body, which could cause problems in early childhood neurodevelopment [29,30], behavioral disorders [31], a drop in IQ, etc. [32]. It can cause devastating damage not only to brain development, but also to kidney function [33,34], histopathological changes and oxidative damage [35,36]. 

Copper (Cu) is also toxic when its level is greater than 0.01 mg/L. The main form of Cu in water is the hydrated ion Cu^2+^ or [Cu(H_2_O)_6_]^2+^, and it is also able to form [CuCl_4_]^2−^ according to the difference in anions. Cu(II) is widely used in agriculture as a pesticide and fungicide, resulting in its ability to easily pollute underground rivers and irrigation lakes, but also its ability to easily harm the ecological environment due to the food chain and enrichment [37]. Excessive copper compounds can lead to soil flora disturbance [38], weight loss and impairment of the reproductive activity of earthworms [39], poor growth of crops and the contamination of grains. In the human body, when the intake of copper exceeds the processing capacity of the human liver, the liver will release copper directly into the blood, where it bonds to the sulfhydryl group of the red blood cell membrane and inhibits G6PD activity, resulting in damage to hemoglobin and red blood cell membrane. In addition, copper can oxidize the fat of lysosomes, resulting in the rupture of lysosome membranes and the release of a large amount of hydrolase, which would cause liver tissue necrosis. Cu^2+^ ions can inhibit cell viability and alter the mRNA expression of cell cycle-related genes and the production of antioxidant oxidation [40], resulting in kidney damage, gastrointestinal distress [41], anemia, coma, and eventually death [42]. 

### 2.2. Characteristics of Wastewater Containing Pb(II) and Cu(II)

Various wastewaters containing Pb(II) and Cu(II) ions are discharged from polluted mining areas, industrial wastewater and urban and agricultural areas. According to the characteristics of the environment, the contents and forms of Pb(II) and Cu(II) in sewage varies in different areas, among which the industrial effluent and domestic sewage discharge as the major sources are mostly caused by insufficient and limited water purification treatment [18]. Ogamba et al. [15] reported that the lead and copper concentrations in the surface water of Taylor creek, Bayelsa State, Nigeria, were 0.00–0.48 and 0.05–0.61 mg/L, respectively. Paul et al. [16] studied the distribution of heavy metal ions within different regions of the Ganges River and found that their concentrations in water from some regions were far above the acceptable levels. In Rishikesh-Allahabad, Pb(II) and Cu(II) concentrations reached 0–36,000 and 2400–26,900 μg/L, respectively. In the Yamuna River, Pb(II) and Cu(II) concentrations reached 12.09–23.31 and 12.01–19.36 mg/L, respectively. Similar pollution conditions were also found by Kumar et al. in the Harike wetland and the Sutlej River [17]. Drozdova [43] investigated the distribution and concentration of heavy metal ions in wastewater from different functional areas in the city of Ostrava, Czech Republic. The results showed that the highest concentrations of Pb(II) and Cu(II) were 26.0 ± 9.3 and 62.3 ± 18.6 μg/L, respectively, indicating that pollution in domestic wastewater in densely populated areas of cities is sometimes more important than industrial wastewater. 

Pb(II) and Cu(II) can also contaminate groundwater through rivers, lakes and sewers. Groundwater is often used for irrigation, which can lead to direct contamination of soil and crops [44]. Jaboobi [45] investigated heavy metal contamination in shallow wells using wastewater, soil and vegetables in Morocco. The results showed that the contents of Pb(II) and Cu(II) in shallow wells were 0.014 and 0.062 mg/L, respectively, and the concentrations of heavy metals in canal wastewater were significantly higher than the recommended values of the FAO and WHO, with average values of 0.043 and 0.093 mg/L, respectively. It is worth noting that the concentrations of Pb(II) and Cu(II) in vegetable test samples were 13.78 and 16.41 mg/kg, exceeding the allowable content, which indicated that wastewater used for irrigation would lead to excessive heavy metal content in crops and indirectly harm the human body. 

## 3. Methods for the Treatment of Heavy Metal Pollution

In view of the current industrial production process, many methods are applied to reduce the concentration of heavy metal ions in sewage, such as the electrochemical method, the membrane filtration method, the chemical precipitation method, the ion exchange method, the adsorption method, etc. At present, each method has its advantages and disadvantages. In practical applications, in the face of complex and changeable pollution and water conditions, it is usually necessary to use a variety of methods to enhance water pollution treatment capacity.

The electrochemical method is an environmentally friendly technology that converts heavy metal ions into precipitation and gas after a redox reaction through the action of an electric current, which removes multiple pollutants at the same time and does not introduce secondary pollution [46]. However, the high energy consumption, the frequent replacement of the electrode material and the drawback of complex pollutants limits the application of the electrochemical method. 

Membrane filtration technology has the advantages of high removal efficiency, simple operation, low energy consumption and environmental friendliness [47]. However, the traditional polymer separation membranes are easily contaminated and have limited stability, which hinders the wide application of membrane filtration.

The chemical precipitation method is a traditional method with the advantages of simple operational, high efficiency and low cost [48]. The precipitating agents such as hydroxide, sulfide and barium salt, can convert soluble heavy metal ions into insoluble sediments in water. However, secondary pollution of the precipitant is the main problem. 

The ion exchange method refers to the exchange of ions in the water body with ions on a certain ion exchanger to achieve the purpose of removing heavy metal ions in the solution [49]. Because the ion exchange method involves the diffusion process, most of the ion exchange process is reversible and has strong recyclability. However, the speed of the ion exchange method in removing heavy metal ions is slow, and it is difficult to use to treat sewage in large quantities. 

The adsorption method is easily operational and low-cost, with a low residue of heavy metal ions [32]. According to the nature of the interaction between the adsorbent surface and the adsorbate, adsorption can be divided into physical adsorption and chemical adsorption, the cause of which can be divided into dispersion force, dipole interaction, quadrupole interaction, electrostatic force, charge transfer interaction, surface modification and pore adsorption. Physical adsorption is mainly generated by intermolecular attraction, which is weak and has low selectivity. Electrostatic attraction is an important mechanism in physical adsorption. Since heavy metals are mostly in the form of positively charged ions in water bodies, the surface electrical properties of adsorbed materials may be conducive to the accumulation and adsorption of heavy metal ions. Chemical adsorption involves the formation of chemical bonds between the adsorbent and adsorbate, which are stronger and more selective. The complexation reaction is the most important component in the chemisorption mechanism of heavy metal ions. It refers to the way by which the organic groups on the surface of the adsorption material are bound to heavy metal ions to form complexes. These groups include amino and sulfhydryl groups, as well as hydroxyl, carboxyl, carbonyl and other oxygen-containing groups. Organic compounds containing the above groups are often used for the surface modification of adsorption materials. A good adsorbent for heavy metal adsorption usually has the following characteristics: large specific surface area and pore volume, suitable pore structure, high selectivity and would not react with the medium and other substances in the application environment to cause secondary pollution. However, it might also be influenced by parameters such as the number of active sites, temperature, pH, the number of adsorbents and adsorbate/competing ions, contact time and the detailed mixing methods.

## 4. Research Progress on Pb(II) and Cu(II) Adsorption

During the past decades, various nanomaterials have been reported for the adsorption of heavy metals, such as carbonaceous materials [50,51], MOFs [52], clay [53], nano zero-valent iron (nZVI) [54] and other nanocomposite materials. Carbonaceous adsorbents include activated carbon (including low-cost biomass-based activated carbon), carbon nanotubes, graphene, graphene oxides (GOs), modified GOs, GO-based nanocomposite, charcoal, plant ash, furnace ash, etc. Clay adsorbents include montmorillonite, kaolinite, zeolite, diatomite, sepiolite, halloysite and various modified clay materials in combination with organic components of resin sorbents, chitosan sorbents, etc. Various types of nZVI and their derivatives, with excellent recyclable abilities, have also been selected. 

Table 2 and Table 3 list several typical adsorption materials and some superior nanomaterials reported recently for the efficient adsorption of Pb(II) and Cu(II) in water. The surface area (SSA), pH, adsorption capacity, selectivity, cost, regeneration ability and recyclable ability of the nanomaterials are summarized. It is found that only few nanomaterials exhibit good selectivity and regeneration ability, and the synthesis cost for most of them is high. The addition of magnetic components, such as Fe or Fe_3_O_4_, makes some of the nanomaterials recyclable. And the microstructures of some novel adsorption materials with outstanding adsorption capacity for Pb(II) and Cu(II) are given in Figure 1 and Figure 2, respectively.

**Table 2 nanomaterials-14-01037-t002:** Adsorption performances of different types of adsorption materials for Pb(II).

Adsorbent Type	Sample	SSA (m^2^/g)	pH	Adsorption Capacity (mg/g)	Selectivity	Cost	Regeneration	Recyclable	Ref.
Activated carbon	Poultry litter-based activated carbon	403	5.0	195.804	/	low	/	/	[55]
	Eucalyptus bark-based activated carbon	1239.38	5.0	109.71	/	low	/	/	[56]
	Magnetized activated carbons	699.9	6	253.2	low	low	/	Yes	[57]
	Carbon foam	458.59	7	491	/	low	/	/	[58]
	Amine-functionalized nano-porous carbon	157	6–8	161.41	low	low	low	/	[59]
	Magnetic pomelo peel biochar	/	6	205.39	high	low	/	Yes	[60]
CNTs	CNTs	/	5	102.04	/	high	/	/	[61]
	Multiwall CNTs	/	3	8118	/	high	/	/	[62]
	CNT–steel slag composite	49.85	6.5	427.26	/	high	/	/	[63]
Graphene	Graphene nanosheets	1000	4	22.42	/	high	/	/	[64]
GO	GO	/	5	1119	/	high	/	/	[65]
	GO	/	/	120	/	high	/	/	[66]
	FGO	120	7	1850	/	high	/	/	[67]
Modified GO	EDTA-GO	623	6.8	479	/	high	/	/	[68]
	Polyethyleneimine-grafted GO	/	6	64.94	/	high	/	/	[69]
	Dipyridylamine-GO	/	4.94	369.749	/	high	high	/	[70]
	rGO/Poly(Acrylamide)	/	6	1000	/	high	/	/	[71]
GO-based nanocomposite	GO-MnFe_2_O_4_	196	5	673	/	high	/	Yes	[72]
	3D graphene/δ-MnO_2_	/	6	643.62	/	high	high	/	[73]
	Fe_3_O_4_/SiO_2_-GO	/	7	385.1	/	high	high	Yes	[74]
Clay	Montmorillonite	/	5.7	31.1	/	low	/	/	[75]
	Kaolinite	18.4	5	31.75	/	low	/	/	[76]
	Natural kaolin	7.98	5	165.117	/	low	/	/	[77]
	Kaolinite	/	5.5	2.37	/	low	/	/	[78]
	Kaolinite	3.7	6.5	4.2	/	low	/	/	[79]
	Sodic montmorillonite	93	5	68.5	/	low	/	/	[80]
	Zeolite	16.3	6	106.61	/	low	/	/	[81]
	Zeolite	473.54	6	15.96	/	low	high	/	[82]
	L-lysine-modified montmorillonite	/	5.5	89.72	/	low	/	/	[83]
Modified clay	Modified kaolinite	10.2	6	20	/	low	/	/	[79]
	Amino-modified attapulgite	/	6	50.66	/	low	/	/	[84]
Clay-based nanocomposite	Fe-Mg LDH@bentonite Surface	154.83	7	1215.81	high	low	/	/	[85]
	Polyamide-amine–magnetic halloysite nanotubes	/	5.6	194.4	/	high	high	/	[86]
	Magnetic halloysite nanotubes/MnO_2_	/	6	59.9	/	high	high	Yes	[50]
	Halloysite/Fe_3_O_4_/polyethylene oxide/chitosan	38.23	5	160	/	high	high	Yes	[87]
	Titanium hydroxyl-grafted silica nanosheets	259	5.8–6.0	184	high	high	high	/	[88]
	SiO_2_/kaolinite/Fe_2_O_3_	/	6	166.67	/	low	/	/	[89]
	Tourmaline–montmorillonite composite	/	6	303.21	high	low	/	/	[90]
	Aminopropyltriethoxysilane-modified magnetic attapulgite@chitosan	/	6	625.34	/	low	high	Yes	[91]
Nano zero-valent iron	g-C_3_N_4_-nZVI	/	5.5	400	/	high	high	Yes	[54]
	nZVI	/	7	119	/	high	/	Yes	[92]
	nZVI	/	6	199	/	high	/	Yes	[93]
	undried nZVI	/	6	807.23	/	high	/	Yes	[94]
ZVI nanocomposite	Phosphoric titanium dioxide-3nZVI	/	6	303.03	/	high	/	Yes	[95]
	Zeolite-supported nZVI	/	6.5	85.37	/	high	/	Yes	[96]
Other nanocomposite	Fe_3_O_4_-FeMoS_4_	/	5	190.75	/	high	high	Yes	[97]
	Dithiocarbamate chitosan@sewage sludge-derived biochar	53.16	5.5	228.69	/	high	high	/	[98]
	Fe_3_O_4_@3-aminopropyltriethoxysilane @ acrylic acid–co-crotonic acid	/	6	78.8	/	high	high	Yes	[99]
	Mg/Fe LDH with Fe_3_O_4_-carbon spheres	4.38	7	696.19	/	high	high	Yes	[100]
	MoS_2_ nanosheets.	/	6	740	high	high	high	/	[101]
	Chitosan–cellulose-Fe(III)	/	4	99.86	/	high	high	Yes	[8]
	PVA/chitosan nanofibers membranes	/	6	266.12	high	high	/	Yes	[102]
	2-aminoterephtalic acid-modified Fe_3_O_4_@triamine-triethoxysilane	114	5.7	205.2	/	high	high	Yes	[103]
	Titanate nanotubes	272.31	5	546.48	high	high	high	/	[104]
MOFs	Zr-MOF	42.9	4	273.2	high	high	high	/	[105]
	Amino-citric anhydride-MIL-53	/	5.8	390	/	high	/	/	[106]
	MOF-808-EDTA	1173		313	/	high	/	/	[52]
	ZIF-67	1289	6	1348.42	/	high	/	/	[107]
	MOF-545	2192	7	73	high	high	high	/	[108]
	Magnetic cellulose nanocrystal/Zn-BTC	65.10	5.45	558.66	high	high	high	Yes	[109]
	Cu-MOFs/Fe_3_O_4_	35.4	/	219.00	/	high	high	Yes	[110]
	Melamine-MOFs	371	5	122	/	high	high	/	[111]
	Thiourea-modified UiO-66-NH_2_	470	/	232	/	high	/	/	[112]
	ZIF-60	/	4	1905	/	high	/	/	[32]

**Table 3 nanomaterials-14-01037-t003:** Adsorption performances of different types of adsorption materials for Cu(II).

Adsorbent Type	Sample	SSA (m^2^/g)	pH	Adsorption Capacity (mg/g)	Selectivity	Cost	Regeneration	Recyclable	Ref.
Activated carbon	Palm shell activated carbon		6.3	30.72	/	low	/	/	[113]
	Chestnut shell activated carbon	1319	5	100	/	low	/	/	[114]
	Grapeseed activated carbon	916	5	48.78	/	low	/	/	[114]
	Eucalyptus bark-based activated carbon	1239.38	5.0	27	/	low	/	/	[56]
	Carbon foam	458.59	7	247	/	low	/	/	[58]
	Porous carbon	157	7–8	46.88	/	low	/	/	[59]
	Magnetic pomelo peel biochar	/	6	81.91	high	low	/	Yes	[60]
	Activated carbon-Na	786	5	17.80	/	low	/	/	[115]
	Banana straw biochar	13.30	6.5	66.23	/	low	/	/	[116]
	Hydroxyapatite-sludge-based biochar	/	6	89.98	/	low	/	/	[117]
CNTs	As-produced CNTs	82.2	6	8.25	/	/	/	/	[118]
	Multiwalled carbon nanotubes	/	5	24.49	/	/	/	/	[119]
	NaOCl-modified CNTs	94.9	6	47.39	/	/	/	/	[118]
	CNTs–steel slag composite	49.85	6.5	132.79	/	/	/	/	[63]
GO	GO	/	5	46.6	/	/	/	/	[120]
	GO	/	3–7	294	/	/	/	/	[65]
	GO	/	5.3	117.5	/	/	/	/	[121]
GO-based nanocomposite	3D graphene/δ-MnO_2_	/	6	228.46	/	/	high	/	[73]
Modified GO	Amino-modified magnetic GO with polyamidoamine dendrimer	/	7.2	353.59	/	/	/	/	[122]
	Magnetic chitosan–GO	132.9	8	217.4	/	/	/	Yes	[123]
	Dipyridylamine-GO	/	5	358.824	/	/	/	/	[70]
	PVP-rGO	/	3.5	1689	/	/	/	/	[124]
Peat	Peat		5	14.3	/	low	/	/	[125]
Fly ash	Mesoporous aluminosilicate	704	4.4	221	/	low	/	/	[126]
Clay	kaolinite	/	6	10.787	/	low	/	/	[127]
	kaolinite	12.57	5	44.66	/	low	/	/	[128]
	Modified kaolinite	32.91	6.5–7.0	1.16	/	low	/	/	[129]
	Natural bentonite	/	6	32.26		low	high	/	[130]
	Na-montmorillonite	/	5.6	33.3	/	low	high	/	[131]
Modified Clay	Bentonite-NH_2_	27.1	5–6	45.8	/	low	high	/	[132]
	Bentonite-COOH	25.5	5–6	53.1	/	low	high	/	[132]
	Polyethylene oxide–chitosan–magnetic halloysite nanotubes	38.23	7	150	/	low	high	Yes	[87]
	Acid-activated montmorillonite-illite	251	4.15	26.09	/	low	/	/	[133]
Clay-based nanocomposite	SiO_2_/kaolinite/Fe_2_O_3_	/	6	153.85	/	low	/	/	[89]
	TiO_2_–acid-activated kaolinite	32.98	7	0.169	/	low	/	/	[134]
Other Nanocomposite	MCs@Mg/Fe-LDHs	4.38	6.5	341.12	high	/	high	Yes	[100]
	Goethite	71.49	5.2	149.25	/	low	/	Yes	[135]
	Hematite	24.82	5.2	84.46	/	low	/	Yes	[135]
	Fe_3_O_4_-FeMoS_4_	/	5	110	/	/	high	Yes	[97]
	Nanohydrated zirconium oxide in polymer exchangers D201	/	7	130	high	/	high		[136]
	Magnetic ferrite nanoparticles	26.78	8	124.80	/	low	/	Yes	[137]
	Zwitterion–chitosan bed	/	6	123.50	/	low	high	/	[138]
	Hollow Fe_3_O_4_@polydopamine	39.96	8	86.35	/	/	high	Yes	[139]
	Melamine-based dendrimer amines–SBA-15	293	5	126.2	/	/	/	/	[140]
	Mesoporous silica	/	5.2	182.39	/	/	/	Yes	[141]
	Titanate nanotubes	272.31	6	122.88	high	low	high	/	[104]
Biomaterial	Tetrazole-bonded bagasse	/	7	132.5	high	low	high	/	[142]
	Caulerpa lentillifera	0.044	6	5.61	/	low	/	/	[143]
	Hydroclathrus clathratus	/	6.2	43.4	/	low	/	/	[144]
	Rosa petal waste biomass	/	5	52.84	/	low	/	/	[145]
MOFs	Fe_3_O_4_@ZIF-8	724.7	6	301.33	high	/	high	Yes	[146]

**Figure 1 nanomaterials-14-01037-f001:**
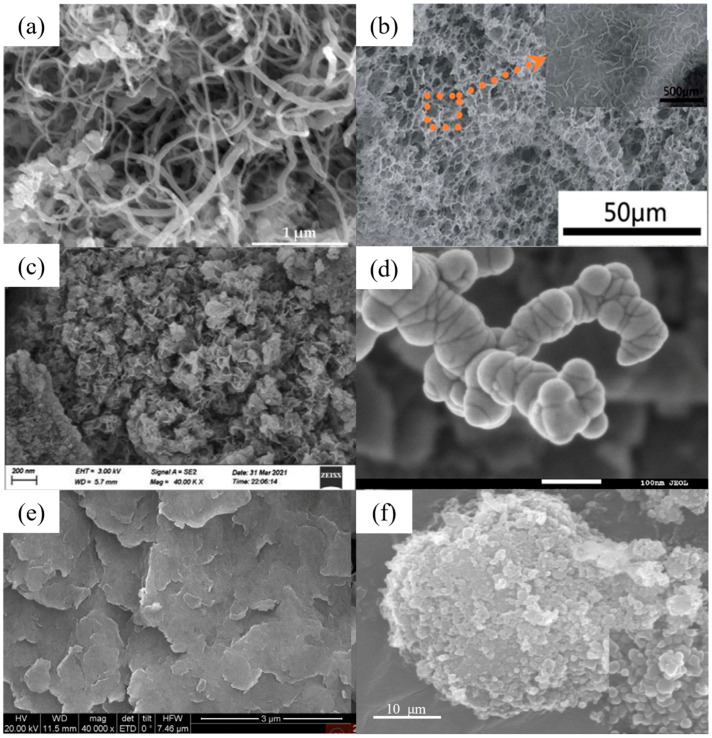
SEM images of (**a**) CNTs synthesized for 45 min (reproduced with permission from Ref. [63] from MDPI). (**b**) Three-dimensional graphene–MnO_2_ (reproduced with permission from Ref. [73] from Elsevier), (**c**) FeMg-LDH@ bentonite (reproduced with permission from Ref. [85] from Elsevier), (**d**) nZVI (reproduced with permission from Ref. [94] from Elsevier), (**e**) Mg/Fe LDH with Fe_3_O_4_–carbon spheres (reproduced with permission from Ref. [100] from Elsevier) and (**f**) ZIF-67 (reproduced with permission from Ref. [107] from Elsevier).

**Figure 2 nanomaterials-14-01037-f002:**
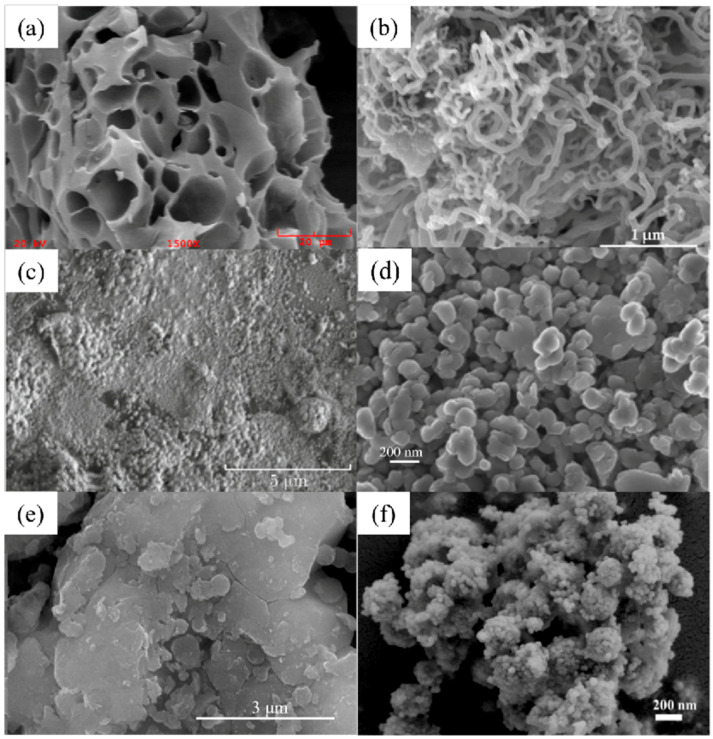
SEM images of (**a**) chestnut shell activated carbon (reproduced with permission from Ref. [114] from Elsevier), (**b**) CNTs (reproduced with permission from Ref. [63] from MDPI), (**c**) PVP-rGO (reproduced with permission from Ref. [124] from Springer), (**d**) TiO_2_–acid-activated kaolinite (reproduced with permission from Ref. [134] from Springer), (**e**) Mg/Fe LDH with Fe_3_O_4_–carbon spheres (reproduced with permission from Ref. [100] from Elsevier) and (**f**) Fe_3_O_4_@ZIF-8 (reproduced with permission from Ref. [146] from Elsevier).

Selectivity is an important factor to evaluate whether an adsorbent will function well in real water systems, which is tested by adding high concentrations of other common metal ions in the solution. Usually, an adsorbent with physical bonds to the metal ions lacks selectivity while an adsorbent with chemical bonds to the metal ions possesses certain selectivity via orbital couplings. Normally, bonding to heavy metal ions via specific groups and particles is the main source for the good selectivity of adsorbent materials, such as -OH [85,104], -TiOH [88], -S [101], Fe_3_O_4_ [146], -COOH [100], etc. In addition, for the ion exchange process, the ionic electricity price, radius, hydration energy and other factors will also affect the selectivity ability of the nanomaterials [85]. Figure 3 lists some novel adsorption materials that have been reported with good selectivity for Pb(II) and Cu(II).

### 4.1. The Influence of Various Parameters on Adsorption Properties

The adsorption properties of adsorbent materials for Pb(II) and Cu(II) are usually affected by pH, temperature, initial concentration, environmental organic matter content and other factors [147]. 

Most adsorbent materials exhibit very low adsorption capacities at low pH due to the ion exchange of H^+^ with the adsorbed ions. Generally, the adsorption performance increases rapidly with the increase in pH at acid conditions (pH < 5), and the increase rate slows down and converges at pH = 5–7, due to the weakened competitiveness of H^+^ and the precipitation of heavy metal ions at higher pH [68,83,95,103,106,116]. For example, the adsorption capacity of EDTA-grafted graphene oxides (EDTA-GO) increased slowly at pH = 2–4, rapidly at pH = 4–6, and reached its maximum at pH = 6–8 for the adsorption of Pb(II) [68]. Similarly, the adsorption capacity of zero-valent iron–phosphate–titanium dioxide (PTO-nZVI) for Pb(II) was negligible at pH = 2, but it increased rapidly at pH = 2–4, and reached its maximum at pH = 4–7 [95]. Although, for the adsorption in alkali conditions (pH > 7), the adsorption capacities become extremely large but this is not so meaningful, since Pb mostly precipitates to form Pb(OH)^+^ or Pb(OH)_2_ at pH > 8 and Cu mostly precipitates to form Cu(OH)^+^ or Cu(OH)_2_ at pH > 9.

The adsorption of Pb(II) and Cu(II) for most of the adsorption materials is an endothermic reaction due to the formation of physical and chemical bonds, and normally the adsorption performance will increase with the temperature [83,114]. In the range of low temperature, temperature has little effect on the adsorption capacity, which is a secondary factor affecting the adsorption capacity. But when the temperature is too high, it may destroy the structure of the adsorption material, resulting in the reduction in adsorption capacity. Zhu et al. [83] investigated the adsorption capacity of the clay composite L-Mt for Pb(II) at 25–55 °C and found that it increased slowly with the increase in temperature and the effect was not obvious. Didem [114] investigated the adsorption capacity of activated carbon for Cu(II) at 25 and 45 °C. The maximum adsorption capacity was 98.04 mg/g at 25 °C. When the temperature increased to 45 °C, it rose to 100.00 mg/g. 

The initial concentration of metal ions can also affect the adsorption capacity. Generally speaking, as the concentration of ions in the solution increases, the adsorption capacity will rise and then stabilize, because the high concentration of ions will provide a greater adsorption driving force and dimer structures might be formed, and then the maximum adsorption capacity will be reached after the adsorption site is fully saturated [82,85]. For example, Ibrahim et al. [82] studied the effect of heavy metal ions at different concentrations of 10–80 mg/L on the adsorption efficiency of zeolite. When the initial concentration was 10–50 mg/L, the adsorption capacity increased significantly. At a concentration above 50 mg/L, it remained stable with little change. Guan et al. [85] studied the adsorption capacity of FeMg-LDH@bentonite at a Pb(II) concentration between 75 and 375 mg/L. The outcome is the same, in that the adsorption capacity in the range of low concentrations (75–250 mg/L) increases continuously, and the adsorption capacity in the range of high concentrations (250–375 mg/L) maintains a balance.

In addition, the content of organic matter in the water body usually also affects the adsorption performance. Some organic compounds, such as carboxylic acid and humic acid, can form complexes with Pb(II) and Cu(II), change the existence form of Pb(II) and Cu(II), and seriously reduce the binding efficiency of their adsorption sites on the adsorption materials, thus reducing the adsorption performance [103]. Abdullah [103] studied the effect of humic acid concentration on Fe_3_O_4_@TATS@ATA adsorption of Pb(II), and found that when humic acid concentration increased, it would bind to Pb and inhibit the adsorption of Pb(II), but when humic acid and adsorption material were pre-balanced, it would promote the adsorption of Pb(II) due to partial combination of humic acid and adsorption material.

### 4.2. Regeneration

Regeneration is an important part of evaluating the comprehensive properties of adsorbent materials. Among all adsorption mechanisms, physical adsorption, electrostatic attraction, surface coupling and ion exchange do not destroy the integrity of the adsorption material during the adsorption process, and the process itself is reversible after the desorption of Pb(II) and Cu(II). Acid treatment with various acids, such as HCl [73,132,142], HNO_3_ [87,139] and citric acid [100] have been used for the desorption of metal ions from the adsorption materials. Since the ion exchange rate of H^+^ with the adsorbed ions increases with the concentration of H^+^, the commonly used concentrations are in the range of 0.1–1 mol/L. In acid treatment, alkali metal or alkaline earth metal salt corresponding to acid can also be added to remove the Pb(II) and Cu(II) at adsorption sites [136]. In addition to acid treatment, chelating and alkali treatment have also been used to desorb Pb(II) and Cu(II) from the adsorption material, by the formation of precipitates and complexes via desorption agents such as EDTA, NaOH [146] and KOH [73]. After the above regeneration and purification, the adsorbed material can be reused. In contrast, for the adsorption materials based on precipitation, reduction and other reactive mechanisms, their nanostructures would be damaged during the adsorption process and hard to regenerate. Furthermore, the regeneration ability of the nanomaterials with large numbers of micropore structures (sizes smaller than 2 nm) compared to other nanomaterials with more macropore structures and better connectivity is normally much lower, since the diffusion efficiency is lower and the microporous structures are more easily blocked by chelating complexes and large organic molecules. 

### 4.3. Carbon-Based Nanomaterials

Carbon-based nanomaterials, with large specific surface areas, rich pore structures and abundant active sites, such as activated carbon, CNTs, graphene and porous carbon, are widely used for environmental protection.

Activated carbon has been widely used in industry for the adsorption of heavy metals. Although several methods have been proposed to prepare activated carbon materials with excellent adsorption properties, their production cost still remains high, and the higher the quality of activated carbon, the greater its cost. There are various sources of activated carbon, and the largest proportion of activated carbon is made from biomass. During the past decades, the preparation of biomass-derived activated carbon with a high specific surface area, high stability, and large adsorption capacity using agricultural waste as raw materials have been extensively studied, since it is abundant, low-cost and easy to produce.

The general preparation process for activated carbon is calcination of the organic raw materials under the certain conditions to reduce the non-carbon components, and then activation by physical (using steam or CO_2_) or chemical methods (KOH, K_2_CO_3_, H_2_SO_4_, etc.) to produce a substance with multi-microporous structures and abundant active groups (Figure 4). The adsorption mechanism of activated carbon is mainly physical adsorption via electrostatic forces, and the adsorption capacity of activated carbon for Pb(II) is usually below 100–200 mg/g, and for Cu(II) is usually around 100 mg/g. Didem [114] used chestnut shell and grape seed as raw materials to prepare activated carbon, which exhibited a huge specific surface area of 1319 m^2^/g, and maximum adsorption capacity of 100 mg/g for Cu(II) at pH = 5.

Furthermore, the adsorption capacity of activated carbon can be significantly improved by grafting organic groups, polymers and loaded nanoparticles on activated carbon. The composite of various materials provides more types of adsorption mechanisms for activated carbon. For example, oxygen-containing or amine-containing organic groups can be complexed, and nanoparticles can be reduced and coprecipitated with Pb(II) and Cu(II) [150]. Figure 5 shows the adsorption mechanism of some activated carbon composites. Zhang et al. [57] prepared magnetized activated carbon by activating rape grass meal, conducting pyrolysis at different temperatures, and then magnetizing activated carbon by the hydrothermal method. The maximum adsorption capacity of MAC-300 for Pb(II) was 253.2 mg/g. The adsorption isotherm and kinetics were consistent with the Freundlich model and the pseudo-second-order kinetic model, respectively, which indicated that the adsorption behavior of the adsorbent depended mainly on the non-uniform active points on the surface of the material. The adsorption mechanism includes surface electrostatic attraction, surface complexation and co-precipitation. Kettum [148] prepared functional carbon materials (C-SO_3_H, C-COOH or C-NH_2_) from coconut shell waste. The composite foam has a porous structure and good carbon dispersion. The adsorption properties of copper ions were also studied. It was found that the maximum adsorption capacity of Cu(II) by C-SO_3_H, C-COOH and C-NH_2_ was 56.5 mg/g, 55.7 mg/g and 41.9 mg/g, respectively. 

Recently, Chen et al. synthesized a novel low-cost tourmaline–montmorillonite nanocomposite (TMM) by the vacuum sintering method, which exhibited excellent adsorption capacity of Pb(II) (303.21 mg/g) in a water system [90]. The absorption mechanisms of Pb(II) on TMMs were found to be mainly multi-layer adsorption, with strong chemical bonding to inner sphere complexes on the surface of the TMMs, which were attributed to the polarization of tourmaline, the adsorption of silanol hybridized groups and the electrostatic forces. The TMMs were considered a promising low-cost nanomaterial with an efficient adsorption ability to remove Pb(II) from water systems. 

Carbon nanotubes (CNTs) are one-dimensional materials with special pore structures and a high density of π electrons, which make their adsorption potential significantly higher than other materials with the same size. Alizadeh [62] prepared multi-walled carbon nanotube–polyrhodanine nanocomposites using a one-step chemical oxidation polymerization method. According to the Langmuir isotherm, it is found that the maximum single-layer adsorption capacity for Pb(II) of the material is 8118 mg/g. The adsorption rate of Pb(II) increased with the increase in solution temperature and followed the quasi-second-order kinetics, indicating that the adsorption mechanism may be a chemisorption process. Although the adsorption effect of carbon nanotubes is excellent, the price of carbon nanotubes is expensive, and there are many difficulties that have not been overcome in practical applications.

Graphene is a well-known two-dimensional material with a unique physical structure and large specific surface area (2630 m^2^/g). However, it is more common to use graphene oxide (GO), since traditional graphene has some drawbacks of site exposure and low adsorption ability [152]. Graphene oxide is one of the graphene derivatives, which contains plenty of highly active functional groups (Figure 6), such as hydroxyl (-OH), carboxylic (-COOH), epoxy (-COC-) and other O-containing functional groups [153]. The mechanism of the adsorption of heavy metal on graphene oxide is relatively complex, including physical adsorption, chemical adsorption and other aspects of influence, which are related to the presence of groups on its surface. Oxygen atoms on the functional groups of graphene oxide can share electrons with metal ions to form metal complexes. These properties increase the heavy metal ion adsorption capacity of graphene oxide to 40–300 mg/g. Raghubanshi [66] used “Hummers” and “improved” methods to prepare GO. Its adsorption capacity for Pb(II) is 120 mg/g. The maximum adsorption capacity of GO for Cu(II) prepared by Yang [120], Sitko [65] and Wu [121] was 46.6, 294 and 117.5 mg/g, respectively.

The adsorption mechanism for the adsorption of Pb(II) and Cu(II) by GO is a combination of physical adsorption, chemical adsorption and electrostatic attraction. The adsorption performance of GO is mainly determined by the properties and concentration of its surface functional groups, and increases with the increase in the surface density of surface groups such as carboxylic acid, the hydroxyl group and the carbonyl group [68].

In order to further improve the adsorption capacity of GO, researchers have grafted various functional groups on GO, such as amino [71], polymers [154] and multi-dentate chelating ligands [70], to achieve better adsorption effects than oxy-containing groups. Outstandingly, the adsorption capacity of some types of GO grafted with functional groups can reach thousands of mg/g. Yang et al. [71] prepared a polyacrylamide (PAM) polymer brush (RGO/PAM) on reduced graphene oxide sheets by in situ radical polymerization. The adsorption capacity of Pb(II) by RGO/PAM is up to 1000 mg/g. Mohammed et al. [69] synthesized polyvinylimide grafted graphene oxide (PEI/GO). The maximum adsorption capacity (q_m_) is 64.94 mg/g. Chemisorption is the main method at low concentrations, and physical adsorption is the main method at high concentrations. The adsorption rate is controlled by the boundary layer diffusion step. In addition, isotherm studies confirmed the strong interaction between Pb^2+^ ions and PEI/GO. Clemonne et al. [68] successfully attached chelating groups to the graphene oxide (GO) surface through a silanization reaction between N-(trimethoxy-silyl) ethylenediamine triacetic acid (EDTA-silane) and the hydroxyl group on the GO surface. Due to the chelating ability of EDTA, the adsorption capacity of graphene oxide was improved. When the pH was 6.8, the adsorption capacity of Pb(II) was 479 ± 46 mg/g, and the adsorption process was completed within 20 min. 

Loading functional nanoparticles on GO provides additional functionality, improving reusability, separability and selectivity in the face of more ions (Figure 7). Liu et al. [73] fabricated 3D graphene/δ-MnO_2_ by uniformly depositing a large number of MnO_2_ nanosheets onto a graphene framework. The material has a fast adsorption kinetic rate and excellent adsorption capacity for heavy metal ions. The saturated adsorption capacities of 3D graphene–c aerogel for Pb^2+^, Cd^2+^ and Cu^2+^ was up to 643.62 mg/g, 250.31 mg/g and 228.46 mg/g, respectively, significantly exceeding that of the corresponding original 3D graphene and d-MnO_2_ nanosheets. And the material has good circulation and separation ability.

By combining two composites with high adsorption capacity and reusability, Hossein et al. [123] synthesized a novel magnetic chitosan–graphene oxide nanocomposite (MCGON). The magnetic MCGONs have a high specific surface area (132.9 m^2^/g), large pore volume (4.03 cm^3^/g) and pore size (15 nm) and strong saturation magnetization (3.82 emu/g). The maximum adsorption capacity of Cu^2+^ by MCGON is 217.4 mg/g. Bao et al. [74] used n-propyl trimethoxy silane as a crosslinking agent to connect Fe_3_O_4_/SiO_2_ to prepare a magnetic graphene oxide nanomaterial. The prepared magnetic graphene oxide can be quickly separated from its aqueous solution by permanent magnets, and has excellent adsorption properties for Cd(II) and Pb(II), with a maximum adsorption capacity of 128.2 and 385.1 mg/g, respectively. In addition, the prepared Fe_3_O_4_/SiO_2_-GO adsorbent can be recycled and has good repeatability.

### 4.4. Clay

Clay is generally formed by the weathering of silica aluminate minerals on the earth’s surface and is widely distributed all over the world. The adsorption of heavy metal on clay materials have been widely studied, using 1:1 layered clay (kaolin and halloysite), 2:1 layered clay (montmorillonite, vermiculite, attapulgite and sepiolite) or other clay minerals [155]. The edges and faces of the clays are generally the most favored positions for the adsorption of various species (cations and anions forms of heavy metal contaminants) in water systems. In addition, for montmorillonite and layered double hydroxide (LDH), the cations or anions in the layer spaces are positions for heavy metal adsorption through the ion exchange mechanism. The cation exchange capacity of clay minerals is highly influenced by the pH value of the solution, since their surfaces would be protonated (positive charged) or deprotonated (negative charged) at a pH below or above the Zero Point of Charge. Although the layered structure of clay minerals possesses a large specific surface area, rich hydroxyl content and good ion exchange ability, the adsorption capacity of natural clay is generally poor. For example, the adsorption capacity of kaolinite and montmorillonite for Cu(II) is only 10.8 [127] and 33.3 mg/L [131], respectively. Based on the mineral characteristics described above, the adsorption mechanism for the adsorption of Pb(II) and Cu(II) on clay materials mainly includes ion exchange, physical adsorption, electrostatic attraction, ion chelation and so on [82,85,88]. In recent years, modified clays, using acid/alkali activation, surface modification (such as amino, carboxyl, hydroxylated groups, etc.) [132], intercalation and pillaring methods, have also been widely studied to improve the adsorption capacity and efficiency of clay minerals (Figure 8). Most of the adsorption mechanisms are related to the acid–base, charge transfer, or ion exchange reactions [156]. Xu et al. [84] grafted amino groups on attapulgite and obtained a nanocomposite (M-ATP) with a maximum adsorption capacity of 50.66 mg/L for Pb^2+^ (Figure 8a). Fourier transform infrared spectroscopy (FT-IR) and X-ray powder diffraction (XRD) showed that a new Si-O-Si bond was formed after modification and the grafting reaction took place on Si-O tetrahedral surface. 

Nanoparticle loading is also an efficient approach to prepared novel clay materials with organic/inorganic composite structures [50]. Guan et al. [85] prepared FeMg-LDH-loaded bentonite (FeMg-LDH@bentonite) by an in situ co-precipitation method (Figure 8b). The maximum adsorption capacity of the material for Cd(II) and Pb(II) was found to be as high as 510.2 mg/g and 1397.62 mg/g, respectively, which is much higher than other conventional adsorbents. The adsorption mechanism showed that the high adsorption ability was caused by the surface complexation, ion exchange and chemical deposition between FeMg-LDH@bentonite and heavy metals (Figure 9a–d). Yan et al.[88] prepared a novel TP-SiNSs nanocomposite with plenty of TiOH groups on its surface by combining calcination, acid leaching and Ti(OH)_4_ grafting methods (Figure 8c). The obtained nanomaterial with dispersed TiOH groups on silica layers (Figure 9e–h) exhibited a short adsorption equilibrium time (within 5 min), large adsorption capacity (TP-SiNSs ≈ 38,000 kg per kg of contaminated drinking water, effluent Pb(II) content <10 µg L^−1^) and excellent renewable and selective properties. Liang et al. [91] prepared APTS-Fe_3_O_4_/APT@CS composite hydrogel beads by loading functional nanoparticles and groups on attapulgite, which exhibited a maximum adsorption capacity of 625.34 mg/g for Pb(II).

### 4.5. Nano Zero-Valent Iron (nZVI)

The separation of the adsorbents from the adsorption system in industry equipment is one of the challenges for the recycling and regeneration of the adsorbents, since most of the adsorbents are well dispersed in the water system during the adsorption process. To overcome this challenge, magnetic nanoparticles such as nanosized zero-valent iron (nZVI), iron oxides (hematite, magnetite and maghemite) and mixed spinel ferrites are introduced to overcome the separation problem.

nZVI, with excellent magnetic properties and redox activity, is widely used for the removal of heavy metal pollutants. The main adsorption mechanism of nZVI is precipitation and reduction. nZVI can increase the pH value of water and precipitate heavy metal ions. At the same time, Pb(II) and Cu(II) can receive electrons from nZVI and thus be reduced and precipitated [54,96]. However, several issues, such as strong agglomeration of nZVI in aqueous solutions and its weak stability and low selectivity to target pollutants remain [92,93]. Various methods, such as polymer modification, surfactant modification, and stabilizer modification using porous materials (activated carbon, montmorillonite, zeolite, carbon nanotubes, metal–organic frame materials, etc.) have been studied. Recently, Ren et al. [95] developed zero-valent iron–phosphate–titanium dioxide (PTO-3nZVI and PTO-nZVI) nanocomposites by phosphoric acid treatment and zero-valent iron loading (Figure 10a). The adsorption capacity of PTO-3nZVI and PTO-nZVI for the target heavy metal Cd(II) (308 mg/g for PTO-3nZVI and 206 mg/g for PTO-nZVI) was significantly increased by complexation and co-precipitation. In addition, there is competitive adsorption between Cd(II) and co-existing heavy metal ions Pb(II) and Cu(II), resulting in different adsorption effects of PTO-3nZVI and PTO-nZVI. The adsorption efficiency of PTO-3nZVI is Cu(II) > Pb(II) > Cd(II). The adsorption efficiency of PTO-nZVI was Pb(II) > Cu(II) > Cd(II). Wang et al. [93] compared nano-scale zero-valent iron (nZVI) to lime, the most widely used heavy metal precipitator in laboratory and field experiments to remove Pb(II) and Zn(II). The water chemistry, treatment efficiency and reaction products of the two reagents were compared. The study showed that a moderate solution pH and its seed effect played a crucial role in the production of a high-quality effluent. Stable and lower levels of Pb(II) and Zn(II) can easily be obtained with nZVI due to its versatile properties and good tolerance to influent fluctuations due to its inherent pH stabilization properties. SEM characterization, particle size analysis and static sedimentation experiments show that nZVI can generate large, consolidated solids suitable for gravity separation (Figure 10b). Tang et al. loaded iron nanoparticles on graphitic carbon nitride. Tang et al. [54] used g-C_3_N_4_ as a carrier to distribute, stabilize and change the microstructure of nZVI (Figure 10c). It was found that functional groups containing N effectively trapped the metal cations in water and accelerated the mass and electron transfer from the iron core to the surface metal ions. The adsorption effect of g-C_3_N_4_-nZVI in wastewater treatment was more than twice that of naked nZVI. Li et al. [96] prepared zeolite-supported nano zero-valent iron (zeolite-nZVI) by a simplified liquid-phase reduction method (Figure 10d) to adsorb As(III), Cd(II) and Pb(II) in an aqueous solution and soil. At pH 6, the maximum adsorption capacity of zeolite-nZVI was 11.52 mg for As(III)/g, 48.63 mg for Cd(II)/g and 85.37 mg for Pb(II)/g, which was much higher than that of zeolite. Batch experiments show that the adsorption mechanisms of the selected heavy metals are varied, including electrostatic adsorption, ion exchange, oxidation, reduction, co-precipitation, complexation and so on. Due to the formation of polyphase compounds on zeolite-nZVI, synergy and competition between heavy metals occur simultaneously. After mixing with 30 g/kg zeolite-nZVI, most of the arsenic, cadmium and lead in the soil sample was immobilized. 

### 4.6. Nanocomposite

Nanocomposite materials are based on a matrix of polymers with plenty of functional groups (resin, rubber, etc.), which are normally loaded by metal, oxides and other inorganic particles in nano-sizes (Figure 11 and Figure 12). The nanometer size effect of the dispersed phase, large specific surface area, strong interface interaction and unique physical and chemical properties of nanocomposite materials promises the nanocomposite materials better physical and chemical properties than conventional composite materials. Because of their special composite properties, nanocomposites usually have a variety of adsorption mechanisms, such as ion exchange, surface coupling, physical adsorption, electrostatic attraction and so on [100,103,157].

Alqadami et al. [103] has prepared a novel nanocomposite material (Fe_3_O_4_@TATS@ATA) for the adsorption and removal of Pb(II) ions from aqueous environments. The BET specific surface area, mean pore diameter, pore volume and magnetization saturation of Fe_3_O_4_@TATS@ATA are 114 m^2^/g, 6.4 nm, 0.054 cm^3^/g and 22 emu/g, respectively. The adsorption isotherm data show that the adsorption of Pb(II) by Fe_3_O_4_@TATS@ATA conforms to the Langmuir and Dubinin–Raduskevich isotherm models, and the R^2^ value is greater than 0.9. The maximum adsorption capacity of Pb(II) was 205.2 mg/g. Mardhia et al. [8] investigated the removal of Pb(II) from an aqueous solution by magnetic chitosan/cellulose nanofiber-Fe (III) [M-Ch/CNF-Fe(III)] composites. The results show that M-Ch/CNF-Fe(III) composites are porous materials and have potential as heavy metal adsorbents. The maximum adsorption capacity is 99.86mg/L. Mohammad et al. [102] used electrospinning to prepare polyvinyl alcohol/chitosan (PVA/Chi) nanofiber films with selectivity and high adsorption ability to Pb(II) depending on the acidity of the solution. The results show that under the best conditions, different adsorption kinetic models are used to process the adsorption data, and it is confirmed that only the pseudo-second-order model conforms to the adsorption kinetics of Pb(II) and Cd(II) ions. Similarly, the equilibrium data were consistent with the Langmuir adsorption isotherm model, and the maximum adsorption amounts of Pb(II) and Cd(II) ions were 266.12 and 148.79 mg/g, respectively. Figure 12 shows SEM images of some composites.

There is also much research on Cu(II) adsorption nanocomposites. Aziam [6] prepared alginate–Moroccan clay bio composites. The results show that the adsorption processes of Cu^2+^ and Ni^2+^ metal ions conform to the quasi-second-order kinetic model, and the corresponding rate constants are determined. The maximum adsorption capacity of Ni^2+^ was 370.37 mg/g and the maximum adsorption capacity of Cu^2+^ was 454.54 mg/g. In the binary system, the maximum adsorption capacity of Ni^2+^ was 357.14 mg/g, and the maximum adsorption capacity of Cu^2+^ was 370.37 mg/g. Xu et al. [128] studied chitosan/polyvinyl alcohol/ZnO microspheres (CS/PVA/ZnO) by the batch method. CS/PVA/ZnO has good antibacterial properties and biocompatibility. The effects of solution pH, adsorbent dose, contact time, initial metal ion concentration and temperature on the adsorption were investigated. The results show that the adsorption of Cu(II) by CS/PVA/ZnO is spontaneous and endothermic. At pH 4.5, the maximum adsorption capacity of CS/PVA/ZnO is 90.90 mg/g.

There are also many related studies on the adsorption of Pb(II) and Cu(II) by nano-composites with SiO_2_. Mesoporous SiO_2_ has been the focus of adsorption research in recent years because of its controllable morphology, excellent loading capacity and large specific surface area. Putz [159] studied the synthesis of MCM-41 ordered mesoporous silica, and the adsorption determination of Cu(II) and Pb(II) solutions showed that the material exhibited monolayer surface adsorption properties. Under the condition of pH 5, the adsorption capacity of Cu(II) and Pb(II) was 9.7 mg/g and 18.8 mg/g, respectively.

Silica gel is a hydrated amorphous product of SiO_2_, with a large specific surface area, multi-channel structure and other excellent characteristics. It has good adsorption properties and is also an excellent carrier for organic functional groups and nanoparticles. Li et al. [160] prepared a nitrotriacetic acid-modified silica gel material (NTA-silica gel) as the adsorption material for Pb^2+^ and Cu^2+^. The adsorption capacity was 76.22 mg/g and 63.5mg/g, respectively. It also has good selectivity and reproducibility. Ali et al. [161] prepared modified porous silica gel by the sol–gel method, and investigated its adsorption properties for Pb(II). It was found that the adsorption rate of Pb(II) was 98%, the maximum adsorption capacity was 792 mg/g and it had excellent regeneration performance.

### 4.7. MOFs

MOFs, short for metal–organic frameworks, are a class of crystalline porous materials with a periodic network structure formed by the self-assembly of inorganic metal centers (metal ions or metal clusters) and bridged organic ligands (Figure 13). Compared with other traditional porous materials, MOF materials have great advantages in specific surface area, porosity, designability and variety, which determines the diversity of their functions and their wide range of applications. Due to their high specific surface area and good pore structure, MOF materials can accommodate a large capacity and good selectivity of guest molecules.

Wang et al. [105] modified UiO-66-NH_2_ with Ni_0.6_Fe_2.4_O_4_ and polyethylenimide, and synthesized a new magnetic Zr-MOF Ni_0.6_Fe_2.4_O_4_-UIO-66-PEI for the adsorption of Pb(II) and Cr(VI) in water. The maximum adsorption capacity of the material for Pb(II) at pH 4.0 was 273.2 mg/g. The adsorption rate was fast, and the adsorption equilibrium was reached within 60 min. Huang et al. [107] synthesized two porous adsorbents, ZIF-8 and ZIF-67, and studied their removal effects on Pb(II) in wastewater. The results showed that the saturated adsorption capacity of ZIF-8 and ZIF-67 for Pb^2+^ reached 1119.80 and 1348.42 mg/g, respectively, which was much higher than that of almost all other porous materials. When excessive adsorbents were used to treat wastewater, the removal rates of Pb^2+^ by ZIF-8 and ZIF-67 adsorbents exceeded 99.4%. In addition, the two adsorbents also showed rapid adsorption kinetics, and adsorption equilibrium was reached in just tens of minutes. Alqadam [106] used the post-synthetic modification (PSM) method to anchor limonic anhydride (CA) to NH_2_-MIL-53(Al) by a covalent bond between the NH2-MIL-53(Al) amino (-NH_2_) group and CA carboxyl group. The metal–organic framework (MOF) of mesoporous amide citric anhydride [AMCA-MIL-53(Al)] was obtained. The influence of pH, contact time, Pb(II) concentration, temperature, and dose of adsorbent were studied. The maximum adsorption of Pb(II) on AMCA-MIL-53(Al) was 390 mg/g and the adsorption was governed by the amide and carboxylate groups via coordinate and electrostatic bonds. However, the desorption experiments showed only a 79.5% maximum capacity after one-time recovery by 0.1 M HCl. Similar to zeolite, the low regeneration ability of MOFs are attributed to the residue of heavy metal ions in the intrinsic microstructures in MOFs, as shown from the SEM images of MOFs (Figure 14). 

There are few studies on the use of MOFs for Cu(II) adsorption. Jiang et al. [146] synthesized an Fe_3_O_4_@ZIF-8 core–shell magnetic composite by the solvothermal method and applied it to remove Cu^2+^ from wastewater. Fe_3_O_4_@ZIF-8 microspheres have a stable performance, high specific surface area (724.7 m^2^/g), and a Cu^2+^ absorption rate of 301.33 mg/g. In addition, Fe_3_O_4_@ZIF-8 also has a fast adsorption kinetics for Cu^2+^ in wastewater, and the adsorption equilibrium can be reached in only 60 min. The mechanism studies show that ion exchange and coordination reaction are the main mechanisms of Fe_3_O_4_@ZIF-8 removal of Cu^2+^.

### 4.8. Adsorption Column

In the above adsorption studies, most researchers adopt the static adsorption scheme in the laboratory. Although this method is convenient for researchers to explore various characteristics of adsorbed materials, the ion concentration of the solution decreases in the adsorption experiment, and will eventually reach a static adsorption equilibrium, which will not change with time. This does not match the current industrial wastewater treatment system. To adapt to the practical application in industry, adsorption column schemes have been proposed. 

Figure 15 shows some of the typical experimental setups for the dynamic adsorption column. At present, the adsorption column technology in industry includes batch-type, continuous moving-bed, continuous fixed-bed, continuous fluidized-bed and pulsed fluidized-bed [162]. In these facilities, one or more adsorption materials are loaded in the channel, the polluted solution continues to go through the column, the concentration of the solution ions entering the port is always consistent, a new adsorption equilibrium is constantly established in the adsorption column, and there is no static adsorption equilibrium, so the adsorption column is also called dynamic equilibrium. 

One advantage of the adsorption column is that it avoids the difficulty of recycling the adsorption material, and indirectly improves the utilization rate of the adsorption material. The factors affecting the adsorption efficiency of the adsorption column include adsorbent type, water flow rate, adsorbent loading capacity, bed height, initial solution concentration and so on [163]. 

**Figure 15 nanomaterials-14-01037-f015:**
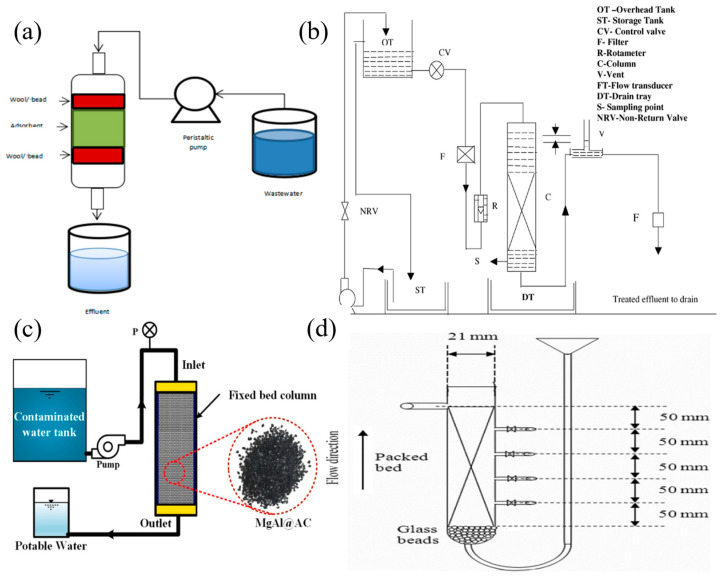
(**a**) Basic design of column adsorption [164]. (**b**) Experimental setup for dynamic column studies [163]. (**c**) Continuous fixed-bed column setup for Pb(II) remediation experiment [165]. (**d**) Schematic diagram of fixed-bed column used in the adsorption study of Pb(II) and Cu(II) ions onto activated carbon and activated carbon–sodium dodecyl sulfate [166].

Many types of adsorption materials have been used in adsorption columns, such as activated carbon [167], graphene [168], clay [169], MOFs [170], etc. Islam et al. [171] synthesized carboxymethylated cellulose as a fixed-bed column adsorption material to adsorb Pb(II) and Cu(II) in single-component and multi-component systems. Under the conditions of pH 5.5, feed flow rate 15 mL/min and feed concentration 10 mg/L (single-component system) and 2.5 mg/L (multi-component system), the removal rate of each metal ion in water by the adsorbent is around 99%. In the single-component system, Pb(II) and Cu(II) interception amounts were 101.0 mg/g and 31.7 mg/g. In the multi-component system, the interception amounts were 20.3 mg/g and 7.8 mg/g, respectively. Olatunji [166] studied the synthesis of adsorption materials by adding sodium dodecyl sulfate (SDS) into coconut shell activated carbon, and investigated its removal rate of Pb(II) or Cu(II) in the resulting wastewater in the adsorption column. It was found that the removal rate of Pb(II) and Cu(II) in wastewater could be significantly increased by adding SDS surfactant on the surface of activated carbon (AC). With the increase in flow rate or initial metal concentration, the breakthrough and failure time is shortened, while with the increase in bed height, the breakthrough and failure time is increased. Zhang et al. [169] prepared a porous silicate/Fe_3_O_4_ adsorbent MCSiMg49, with the adsorption capacity of Pb(II) reaching 499.3 mg/g. The adsorption columns filled with MCSiMg49 (1 g, upper layer) and clay (1 g, lower layer) can completely remove Pb(II) (concentration 50 mg/L) from natural water (such as Yangtze River water or Yellow River water).

## 5. Conclusions

The removal of lead and copper ions has always been a concern of researchers, and the adsorption method is one of the most valuable methods. In recent years, a variety of nanomaterials have been developed, and adsorbents such as MOFs, zero-valent iron, carbonaceous materials, clay materials and nanocomposites have steadily increased for the adsorption of lead and copper ions. Among the various adsorption materials summarized in this review, although the maximum adsorption capacity of CNTs for Pb(II) is as high as 8000 mg/g, and that of modified GO for Cu(II) is around 360 mg/g, their cost and adsorption selectivity still need to be improved. In addition, pH value, temperature, initial metal ion concentration, the amount of adsorbent added and other factors jointly determine adsorption performances. Ion exchange, ion chelation, electrostatic adsorption, precipitation, redox, complex formation, hydrogen bond and other forms of action and forces are the basic mechanisms that relate to the efficient adsorption of lead ions and copper ions in water systems. There are a few points worth noting.

(1) Although various adsorbents with excellent adsorption properties have been reported, the production cost and the probability of the industrial scale-up of the synthesis process need to be considered, as well as the stability and regeneration ability. Most adsorbents were not applied to treat real water samples from industries. The adsorption performance of the adsorbents for contaminated water from polluted water systems should be tested. 

(2) The secondary pollution of the used solid adsorbents with a large amount of hazardous heavy metal has also become a critical environmental problem nowadays. Future works should pay more attention to the selectivity and regeneration of the adsorbents, which determines the cyclic performance and economic potential for the adsorbents for industrial applications. The efficient extraction of the heavy metal content from the used adsorbents would not only provide more secondary resources but also realize the reduction, innocuity and stabilization of the solid hazardous wastes, which could help to finally achieve the target of sludge reduction. 

(3) Although the addition of magnetic components (nZVI or Fe_3_O_4_) makes some of the nanomaterials recyclable, they still faces the drawbacks of chemical instability and weakness in the application to huge water systems (rivers and lakes). Other novel nanomaterials, which can be easily recycled in large systems, might bring more prosperity to adsorption technology. 

## Figures and Tables

**Figure 3 nanomaterials-14-01037-f003:**
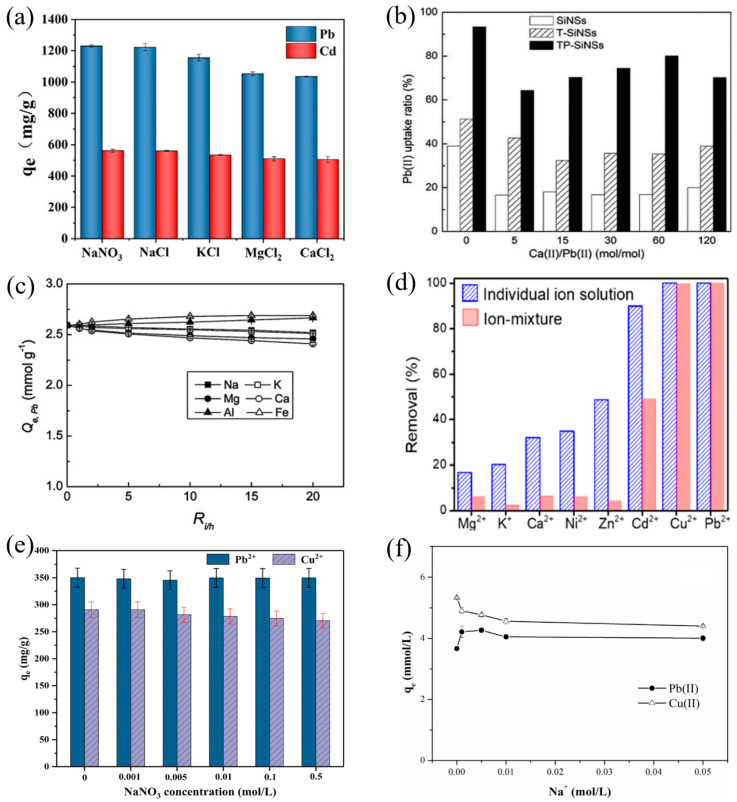
(**a**) FeMg-LDH@bentonite (reproduced with permission from Ref. [85] from Elsevier), (**b**) titanium hydroxyl-grafted silica nanosheets (reproduced with permission from Ref. [88] from Wiley), (**c**) titanate nanotubes (reproduced with permission from Ref. [104] from Elsevier), (**d**) MoS_2_ nanosheets (reproduced with permission from Ref. [101] from ACS), (**e**) Fe_3_O_4_@ZIF-8 (reproduced with permission from Ref. [146] from Elsevier) and (**f**) Mg/Fe LDH with Fe_3_O_4_–carbon spheres (reproduced with permission from Ref. [100] from Elsevier).

**Figure 4 nanomaterials-14-01037-f004:**
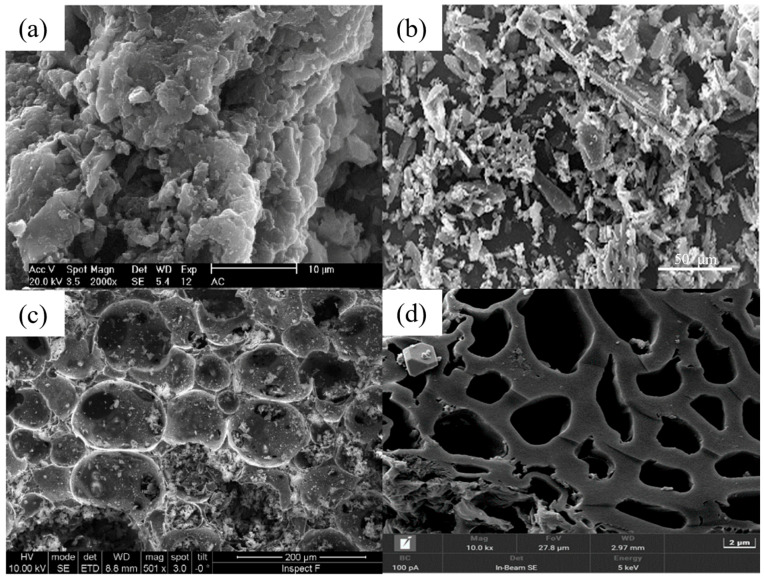
SEM images of (**a**) eucalyptus bark activated carbon (reproduced with permission from Ref. [56] from Elsevier), (**b**) activated carbon (reproduced with permission from Ref. [148] from Nature), (**c**) carbon foam (reproduced with permission from Ref. [58] from Elsevier) and (**d**) magnetic activated carbon (reproduced with permission from Ref. [149] from MDPI).

**Figure 5 nanomaterials-14-01037-f005:**
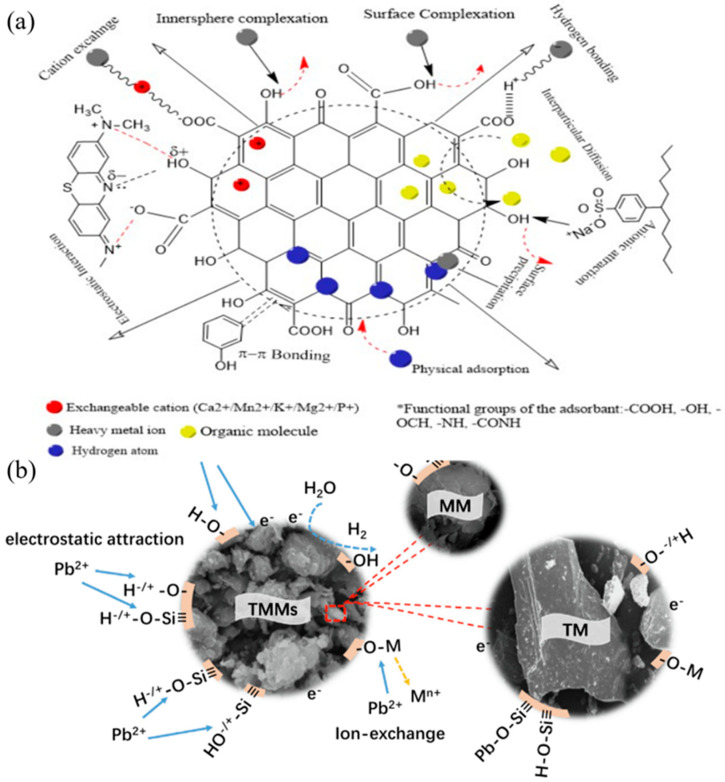
Adsorption mechanism of (**a**) different types of pollutants on the activated carbon surface (reproduced with permission from Ref. [151] from MDPI) and (**b**) tourmaline–montmorillonite composite (reproduced with permission from Ref. [90] from Elsevier).

**Figure 6 nanomaterials-14-01037-f006:**
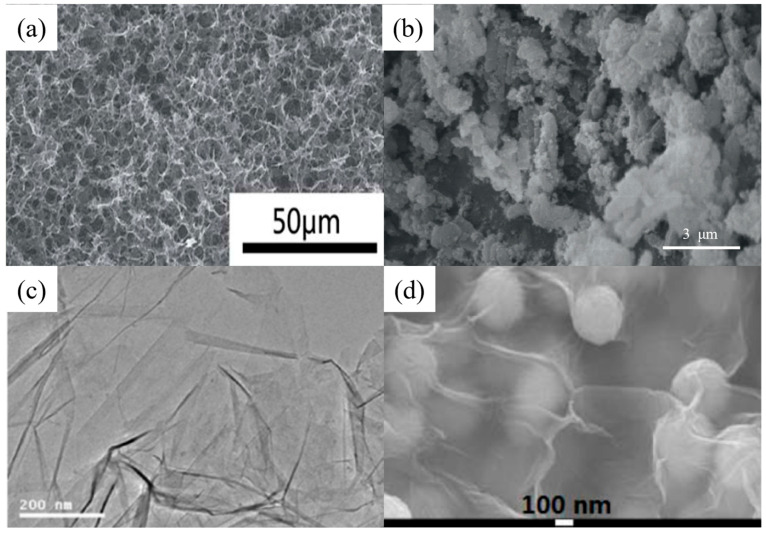
SEM images of (**a**) 3D graphene aerogels (reproduced with permission from Ref. [73] from Elsevier), (**b**) magnetic chitosan–GO (reproduced with permission from Ref. [123] from Elsevier), (**c**) RGO (reproduced with permission from Ref. [71] from ACS) and (**d**) Fe_3_O_4_/SiO_2_-GO (reproduced with permission from Ref. [74] from Elsevier).

**Figure 7 nanomaterials-14-01037-f007:**
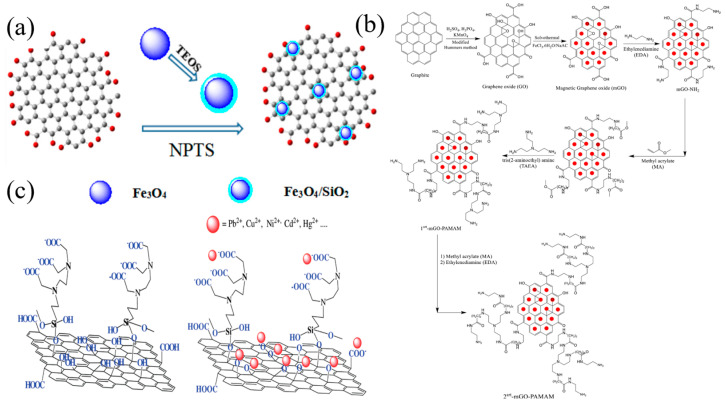
(**a**) Synthesis mechanism of Fe_3_O_4_/SiO_2_-GO (reproduced with permission from Ref. [74] from Elsevier). (**b**) Schematic illustration of the methods used for preparation of amino-modified magnetic GO with polyamidoamine dendrimer (reproduced with permission from Ref. [122] from Elsevier). (**c**) Chemical structure of EDTA-GO (left) and its interaction with heavy metal cations (right) (reproduced with permission from Ref. [68] from ACS).

**Figure 8 nanomaterials-14-01037-f008:**
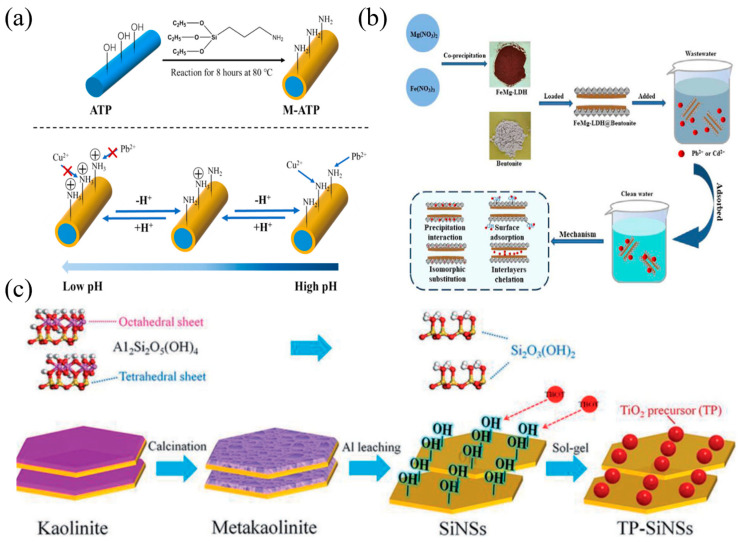
(**a**) Scheme of the modification process of attapulgite and the effect of pH on the adsorption process (reproduced with permission from Ref. [84] from Elsevier). (**b**) Synthesis and adsorption mechanism of FeMg-LDH@bentonite (reproduced with permission from Ref. [85] from Elsevier). (**c**) Schematic illustration for the preparation of titanium hydroxyl-grafted silica nanosheets (reproduced with permission from Ref. [88] from Wiley).

**Figure 9 nanomaterials-14-01037-f009:**
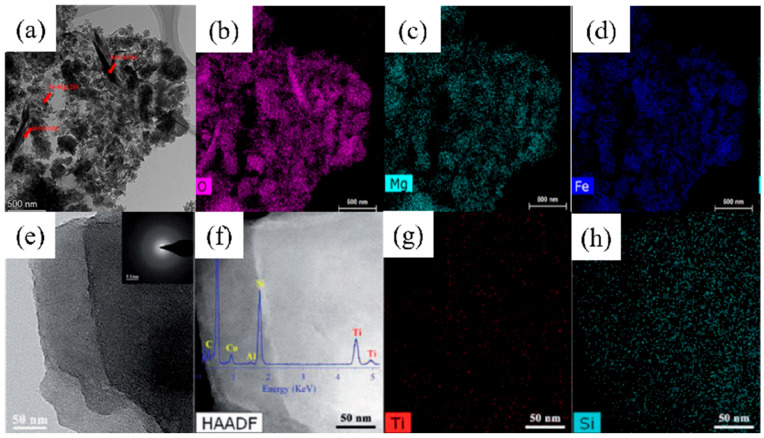
(**a**) TEM images of FeMg-LDH@ bentonite and EDX mapping of (**b**) O element, (**c**) Mg element and (**d**) Fe element (reproduced with permission from Ref. [85] from Elsevier). (**e**) TEM images of titanium hydroxyl-grafted silica nanosheets and (**f**) HAADF-STEM image (inset EDS spectrum), EDS mapping of (**g**) Ti element and (**h**) Si element (reproduced with permission from Ref. [88] from Wiley).

**Figure 10 nanomaterials-14-01037-f010:**
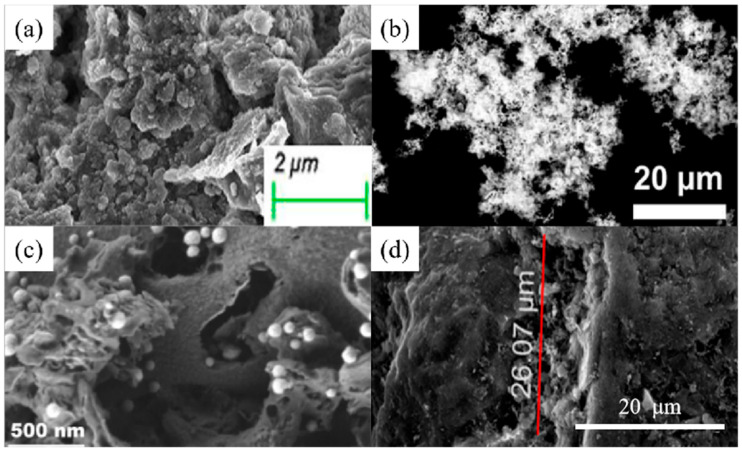
SEM images of (**a**) phosphoric titanium dioxide-3nZVI (reproduced with permission from Ref. [95] from Elsevier), (**b**) nZVI-Pb (reproduced with permission from Ref. [93] from Elsevier), (**c**) g-C_3_N_4_-nZVI (reproduced with permission from Ref. [54] from Elsevier) and (**d**) zeolite-nZVI (reproduced with permission from Ref. [96] from Elsevier).

**Figure 11 nanomaterials-14-01037-f011:**
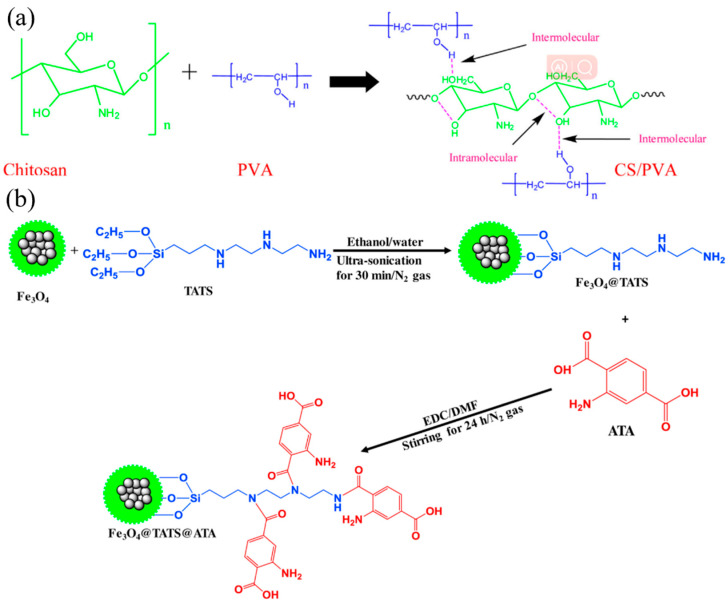
(**a**) Inter- and intramolecular forces between Chitosan and PVA (reproduced with permission from Ref. [158] from Elsevier). (**b**) Preparation of 2-aminoterephtalic acid-modified Fe_3_O_4_@triamine-triethoxysilane nanocomposites (reproduced with permission from Ref. [103] from Elsevier).

**Figure 12 nanomaterials-14-01037-f012:**
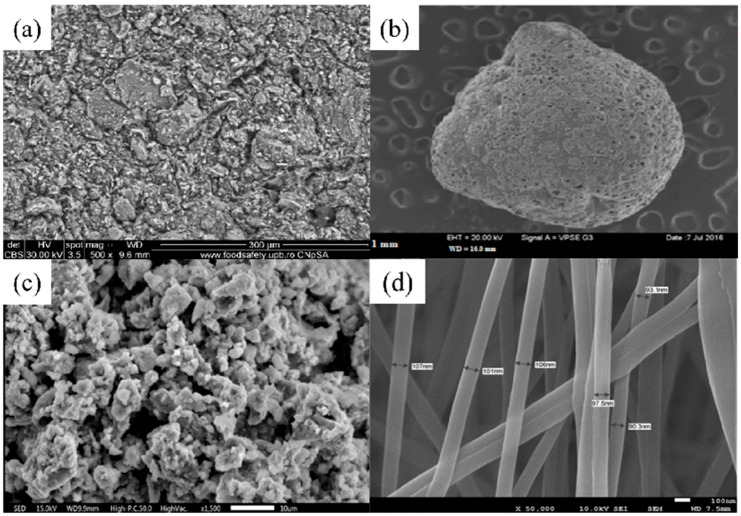
SEM images of (**a**) alginate–Moroccan clay bio-nanocomposite microparticles (reproduced with permission from Ref. [6] from MDPI), (**b**) chitosan/poly(vinylalcohol)/ZnO beads (reproduced with permission from Ref. [128] from Elsevier), (**c**) magnetic chitosan/cellulose nanofiber-Fe(III) composite (reproduced with permission from Ref. [8] from MDPI) and (**d**) FE-SEM images of poly(vinylalcohol)–chitosan nanofibers membranes (reproduced with permission from Ref. [102] from Elsevier).

**Figure 13 nanomaterials-14-01037-f013:**
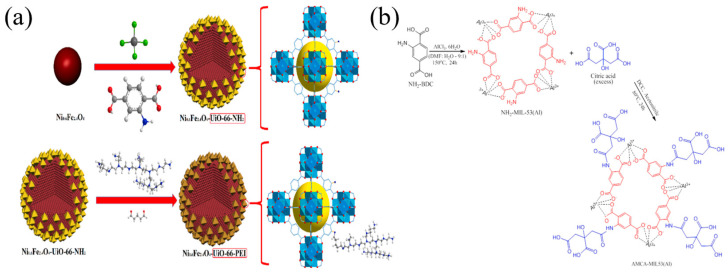
(**a**) A schematic showing the procedures to synthesize Ni_0.6_Fe_2.4_O_4_-UiO-66-types (reproduced with permission from Ref. [105] from Elsevier). (**b**) NH_2_-MIL53(Al) synthesis and post-synthesis modification to amino-citric anhydride-MIL-53 (reproduced with permission from Ref. [106] from Elsevier).

**Figure 14 nanomaterials-14-01037-f014:**
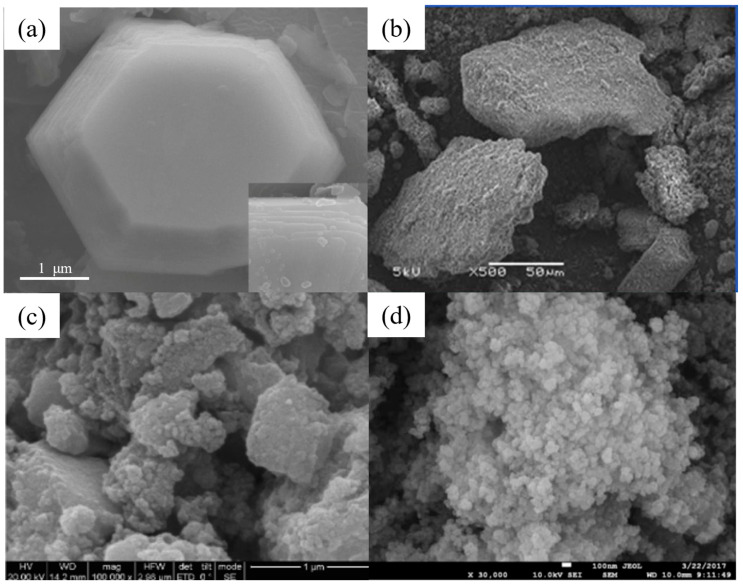
SEM images of (**a**) ZIF-8 (reproduced with permission from Ref. [107] from Elsevier), (**b**) amino-citric anhydride-MIL-53 (reproduced with permission from Ref. [106] from Elsevier), (**c**) Ni_0.6_Fe_2.4_O_4_-PEI (reproduced with permission from Ref. [105] from Elsevier) and (**d**) melamine–MOFs (reproduced with permission from Ref. [111] from Elsevier).

**Table 1 nanomaterials-14-01037-t001:** The sources, allowable concentrations, toxicity, hazards and vulnerable groups for Pb(II) and Cu(II).

Heavy Metal Ion	Source	Allowable Concentration (mg/L)	Toxicity and Hazard	Vulnerable Groups
Pb(II)	Electroplating, mining, paints, batteries, pesticides, coal burning, etc.	0.01 (ISO) 0.01 (WHO) 0.015 (USEPA) 0.01 (MEP)	Gastrointestinal injury, anorexia, anemia, Decreased IQ, loss of appetite, brain damage, malaise, etc.	People with weakened immunity and residents in areas with serious heavy metal pollution
Cu(II)	Paints, electroplating, metallurgical and mining processes, pesticides, alloy manufacturing, etc.	0.05 (ISO) 2.00 (WHO) 1.30 (USEPA) 1.00 (MEP)	Reduced cell viability, kidney damage, anemia, gastrointestinal distress, coma, death	Workers and residents of the industrial zone

ISO: International Organization for Standardization, WHO: World Health Organization, USEPA: United States Environmental Protection Agency, MEP: Ministry of Ecology and Environment of the People’s Republic of China.

## Data Availability

The dataset is available on request from the authors.

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
