# Peer review of "Recent Progress on the Adsorption of Heavy Metal Ions Pb(II) and Cu(II) from Wastewater"

_nanomaterials, 2024, doi:10.3390/nano14121037_

Round 1

Reviewer 1 Report (Previous Reviewer 3)

Comments and Suggestions for Authors

The papers has been seriously revised, and can be recommended for publication after eliminating some errors.

There is no reference in the text to Figure 3

line 400 misprint

lines 404,405 Please avoid repeating excellent and large adsorption capacities

lines 659-660 The sentence should be rephrased: not the adsorbents "do not deal" , but the reported studies did not use/consider/deal/apply real water, or the adsorbents were not applied to real water.

Fullstops have to be deleted after "Figure" in the in-text references to the figures.

ref. 40 Delete fullstops after the first two words in the journal name abbreviation

refs. 45,47 Do not abbreviate Dalton in the journal name.

Author Response

Reviewer 2 Report (Previous Reviewer 2)

Comments and Suggestions for Authors

The authors have done a pretty good job of reworking the review, but I think the manuscript still needs serious revision. And I want to give the authors a chance. As I said last time, the authors should seriously justify the novelty of their review, or the title of the paper should be changed, perhaps to "Resent progress... etc" Why is the review paper devoted specifically to lead and copper ions?

1. It is necessary to separate the introduction and subsections 1.1. and 1.2. by adding to the text the section on the characteristics and toxicological properties of wastewater containing Pb (II) and Cu (II). Thus, it will be 2.1. and 2.2. etc.

2. The introduction must be expanded and the novelty of the review paper must be justified in it.

3. It is necessary to add a separate section about the influence of various parameters (pH, temperature, initial concentration, etc.) on the adsorption process

4. I think that the paper should have a separate section devoted to the regeneration and reuse of sorbents

5) It is necessary to discuss the sorption mechanism in more detail

6) Careful editing is also required. The article should adhere to the same designation "Pb(II) and Cu(II)" or " everywhere. There are a lot of abbreviations in the text of the paper and in the table, which are sometimes very difficult to understand. This makes it difficult to read the paper. I think that all abbreviations should be removed, leaving two or three most frequently used ones.

Author Response

Reviewer 3 Report (New Reviewer)

Comments and Suggestions for Authors

This review paper is well presented. It can be accepted for publication.

Author Response

Thank you for the positive comments.

Round 2

Reviewer 2 Report (Previous Reviewer 2)

Comments and Suggestions for Authors

In general, I agree with the changes made to the manuscript by the authors. However, I still have some minor comments:

1. After you called the manuscript "Recent progress on the adsorption of heavy metal ions Pb(II) and Cu(II) from wastewater", it seems to me that the word "a review" is not needed here.

2. Tables 2, 3 and fugures captions still contains abbreviations that make it difficult to understand. For example, MAC, C-MPG1-micro, MPPB, APTS-Fe3O4/APT@CS etc. there are a lot of them. It seems to me that as few abbreviations as possible should be left, maximum two, three generally accepted.

Author Response

This manuscript is a resubmission of an earlier submission. The following is a list of the peer review reports and author responses from that submission.

Round 1

Reviewer 1 Report

Comments and Suggestions for Authors

I recommend this manuscript for publication after minor revision, which is desired at following points:

Page 2, line 65:

“The main form of Cu in water is the hydrated ion Cu2+ or [Cu(H2O)4]2+, …”

Do you mean the [Cu(H2O)6]2+ ion here?

Page 15, line 302:

“and 33.3mg/L”

Insert a space between number and unit.

The same applies to:

Page 20, line 402: “114m2/g”

Page 21, line 435: “43590.90mg/g”

Page 22, line 449: “273.2mg /g”

Comments on the Quality of English Language

Minor editing of English language required

Reviewer 2 Report

Comments and Suggestions for Authors

The manuscript is dedicated to the review of research on the adsorption of Pb(II) and Cu(II) ions from wastewater. I regret to say that I have to recommend rejecting this manuscript in its current form for publication in the journal. To have the article further considered, I believe a radical overhaul is necessary. Firstly, there have been numerous reviews on this issue recently, some of which include:

- https://doi.org/10.1016/j.heliyon.2023.e15455

- https://doi.org/10.3390/ijerph20053885

- https://doi.org/10.1016/j.chemosphere.2023.138508

- https://doi.org/10.3390/waste1030046

- https://doi.org/10.1007/s40726-023-00290-7

and many others. The authors need to justify how their review significantly differs from those already published.

Secondly, the information provided in lines 49-66 should be expanded, and a separate section on the toxicological effects of Pb(II) and Cu(II) should be included. Additionally, a section should be separately included on the characteristics of wastewater containing Pb(II) and Cu(II), both individually and in conjunction with other heavy metal ions.

Thirdly, I believe it is necessary to discuss the influence of various parameters on the removal of Pb(II) and Cu(II), as well as separately consider the sorption mechanism and kinetics on different sorbents.

Fourthly, attention should be paid to the regeneration of sorbents and the extraction of Pb(II) and Cu(II) from sorbents.

Fifthly, information about implemented wastewater treatment technologies for Pb(II) and Cu(II) should be added, along with a description of adsorption column devices.

Sixthly, it is essential to carefully edit the article, particularly focusing on the figure captions. They should be understandable without referring to the text of the article.

After incorporating all these changes, the article can be reconsidered. Once again, I regret to say that in its current form, I cannot recommend this manuscript for publication due to significant shortcomings.

Reviewer 3 Report

Comments and Suggestions for Authors

This review covers and important topic of water pollution and remediation using sorption technology. Authors reviewed recent publications on various types of materials used as sorbents for heavy metal ions Cu(II) and Pb(II). The review is timely, informative and provides a fair overview on various directions of research in this area. The paper can be recommended for publication, after eliminating a number of deficiencies, as listed below.  

-- line 218: not clear, which metal is referred to.

-- unfinished sentences (line 313, 236,237)

-- One class of popular nanocomposites, silica, silica-inorganic and silica-organic materials are scarcely represented in this review. Some of the recent applications of such materials towards Cu and Pb adsorption can be found, for example, in these papers:

https://doi.org/10.3390/gels8070443
https://doi.org/10.1016/j.ijbiomac.2019.06.244
https://doi.org/10.3390/molecules26226885
https://doi.org/10.1039/C8RA08638A

-- In Conclusions, lines 490-491, the sentence is too general. Authors may write, that the indicated sorption capacities are achieven only among the materials listed in this review.
